# Physical Fitness and Peer Relationships in Spanish Preadolescents

**DOI:** 10.3390/ijerph17061890

**Published:** 2020-03-14

**Authors:** Juan Gregorio Fernández-Bustos, Juan Carlos Pastor-Vicedo, Irene González-Martí, Ricardo Cuevas-Campos

**Affiliations:** 1Faculty of Education, University of Castilla La Mancha, 02071 Albacete, Spain; juang.fernandez@uclm.es (J.G.F.-B.); irene.gmarti@uclm.es (I.G.-M.); 2Faculty of Education, University of Castilla La Mancha, 13001 Ciudad Real, Spain; ricardo.cuevas@uclm.es

**Keywords:** physical fitness, peer relationship, social health, preadolescence

## Abstract

Several studies have linked physical fitness (PF) with improvements in health, at a physiological and psychological level; however, there is little evidence of its relationship with health in the social field. Hence, the main aim of this study was to determine the existing relationship between PF and peer relations, as an indicator of social health in Spanish pre-teens. For that purpose, 642 participants aged 9 to 12 were chosen and given the high-priority Alpha Fitness battery in order to assess the PF, as well as the Classroom Social Experiences Query (CESC) to assess their social status. The results showed that those students with a better cardio-respiratory fitness obtained more nominations from their classmates in pro-sociality and positive status, and fewer in victimisation and negative status. Additionally, although to a lesser extent, muscular fitness was also related to a higher positive status and lower victimisation. The weight condition was also related to social behaviours, although the resulting data were differentiated by sex. While females with a standard weight stood out for their positive status, underweight males obtained worse results in positive status and fewer in negative status. These results all correspond with the aesthetic models imposed by society for females and males, respectively. These results show that PF is related to social health, which makes necessary the promotion of physical activity and the development of PF within the school environment, with attention to its relationship with the social health of the students.

## 1. Introduction

Avoiding unidimensional and exclusively biological conceptions, the concept of health has a complex and multidimensional nature in the individual. We are not only in a physical reality, but we also feel, think, and interact with others. In the last century, the World Health Organization [1] integrated this multidimensional reality by defining health as a complete state of physical, mental, and social well-being, and not only the absence of disease.

The research has proven that physical activity (PA) is one of the most influencing aspects on health, as it may provide not only physiological benefits [2], but also psychological benefits [3] related to mental health [4], including depression, anxiety, executive function [5], psychological well-being [6], body satisfaction [7] or self-concept [8]. Likewise, studies in recent years have reported how PA is also related to cognitive [9] and academic [10] performance. 

In addition to these beneficial effects of PA at the physiological and mental level, there are studies that also relate PA to the social field. One of the indicators used to determine social health has been the peer relationship. Gifford-Smith and Brownell [11] suggested that peer relationships generally contain social status, friendship, and peer nets or groups. Social status mainly refers to the degree that children like their friends, and it is highly related to the children’s satisfaction [12]. Previous studies have claimed that the students’ participation in PA has an influence on the relationships with their peers, as these can be developed within environments that favour practice [13]. 

Lehto, Reunamo, and Ruismäki [14] proved that if participation in PA was lower, contact with other children tended to be weaker, having a higher probability of being included in the category of social rejection. By contrast, active children had more possibilities of developing friendships and being popular among other children. In other words, there is proof that links PA with better socialization [15]. Additionally, considering this relation as bidirectional, it was identified that peers play a significant role in the practice of PA, as the increase of friendship, acceptance, and support among friends promotes the practice [16,17]. It is true that some of these studies specifically focus on sport. Thus, sport has shown to have a positive impact on behaviours such as solidarity, cultural values, and social attitudes and behaviours [18]. Although it is true that sport is not educating and socialising per se, sports contain systematised strategies, which obtain the desired effects [19]. 

Similarly to PA, there is evidence linking physical fitness (PF) with health benefits. The results of the cross-sectional and longitudinal studies in Europe show that PF is a health marker in childhood and the teenage years [20], independent from PA [21,22], and it may be used to forecast the health condition in the later stages of the life of an individual [20]. Therefore, poor PF during childhood and the teenage years is associated with significant health problems, such as a higher risk of being overweight [23], having cardio-vascular diseases [24], skeletal health problems [25], a decrease in life quality [26], and poorer mental health [20]. Furthermore, according to previous researches, PF not only has a positive influence on physical aspects, but also academic and cognitive aspects, having an impact on children’s mental development [27].

There seems to be an agreement on health-related PF having a multidimensional structure with different components [28]. European studies take into account the following: body composition, cardio-respiratory endurance (CR), musculoskeletal endurance, and motor skills (speed, agility, and coordination) [29]. These components, when separate, have shown a close relation between children and teenagers’ health. For instance, important prospective studies have proved unequivocally that CR is the strongest indicator of mortality and morbidity [30]. CR, related to a better cardio-vascular profile during teen years, has also been linked as an indicator of future cardiovascular risk, regardless of weight [20]. Likewise, during childhood, it has been related to better health of the musculoskeletal system and better mental health [31]. Other studies indicated a relation between muscular fitness in the school years and lower cardio-vascular risk, with good psychological health and life satisfaction in children [32], and to better mental health in the adult years [24]. On the other hand, being overweight entails higher cardio-vascular risk and mortality, as well as other health problems linked to it [33], and is related to mental health and self-esteem problems [34].

On the other hand, there is also proof that confirms that children’s health conditions are related to the quality of peer relations [35]. Nevertheless, there are a few studies linking PF with social health aspects, and particularly with peer relations. Chen [36] found a correlation in Asian teenagers between different components of the PF (CR, and muscular fitness) and peer relations, with CR being the aspect with the highest correlation, especially among males. The study of Guillamón and Cantó [37], in children from 8 to 12 years old, showed an association between the muscular strength level and the violence observed, though without documenting personal experience. The most reported topic in the literature is the weight status and peer relation. Different studies agree that overweight students have more problems with their peers [38]. For instance, they present a higher risk of victimisation and being easily rejected [39] and less popular among their peers, obtaining fewer positive nominations [40]. They are even outcast, not only passively by their peers, but also publicly rejected [41].

Therefore, the main aim of this research has been the analysis of the correspondences between different components of the health-oriented PF (CR, body strength and composition), as an indicator of physical health, and peer relations as an indicator of the social dimension of health, especially in the aspects of social status, pro-sociality, victimization, and abuse.

## 2. Materials and Methods 

### 2.1. Design and Participants

We requested the participation of 28 centres in the centre-south area of Spain, and 10 of them willingly showed interest. A total of 642 students of the fifth and sixth grades of primary education (304 boys, 338 girls), from 9 to 12 years old (M = 10.68, SD = 0.91), took part in this cross-sectional study. All participants, and their parents or legal gardians were informed about the aims, procedures, risks, and benefits of the research and were volunteers among those who provided informed consent. The following were considered as rejection criteria: temporary injury or handicap preventing the performance of physical tests, clinical diagnosis of diabetes or non-nutritional diseases, and not providing the informed consent. The study was carried out pursuant to the ethical rules of the Declaration of Helsinki (Hong Kong revision, 1989), the recommendations of Good Clinical Practice of the European Ethics Committees (1 July 1991, document 111/3976/88), and the Spanish legislation on clinical research on humans (Royal Decree 561/1993 on clinical trials)—thus, the anonymity of the participants and the data confidentiality was guaranteed during the whole process. The protocol was approved by the The Ethics Committee on Human Research (Ethic Code: N202001007) (University of Castilla La Mancha). 

### 2.2. Measures and Procedure

#### 2.2.1. Physical Fitness

In order to asses the PF, the high-priority ALPHA health-related fitness test battery was used [29]. This battery version includes the following tests:

Body composition: Body Mass Index (BMI) and waist circumference (WC). BMI from the measurement of weight (Tanita WB 380 S) and size (measuring rod Tanita HR 001) as a reference of the weight condition of the students. In order to set the prevalence of weight and obesity, the cut-offs established by the International Obesity Task Force [42] for age and sex were used, implemented in four categories (underweight, standard weight, overweight, and obese). WC was used to assess the body fat in the abdominal, core, or central area, using an inextensible tape measure.

Musculoskeletal fitness: handgrip strength test (HS) and standing broad jump (SBJ). The HS (digital dynamometre Takei TKK 5401, range 5–100 kg) assesses the maximal handgrip strength. The best result of two attempts with both hands was recorded. SBJ measures the explosive force of the lower body after a jump without any movement forward (cm). The best result of two attempts was recorded.

CR: The 20 m shuttle run test (Course Navette). This test assesses the maximal aerobic endurance in students. The test was performed according to the protocol of de Léger et al. [43]. 

The tests were carried out during two sessions of Physical Education and following the recommended sequence: 1. Weight and Height; 2. WC; 3. HS and SBJ; 4. 20 m shuttle run test. The data recording was carried out by two researchers trained so as to guarantee the measurements’ standardisation and liability, and according to the protocol set out in the instructions manual for the implementation of the ALPHA Fitness tests (http://www.ugr.es/~cts262/ES/documents/MANUALALPHA-Fitness.pdf). 

We used the normative values for physical fitness (20 m shuttle run, SBJ, HS) in European children (aged 9–12 years), in order to classify the results according to sex and age, and this was expressed in percentiles from 10 to 100 [44].

In a study by Ortega et al. [23], the percentiles were used to intuitively classify students according to their level of fitness in five groups: very poor (X < P20), poor (P20 ≤ X < P40), medium (P40 ≤ X < P60), good (P60 ≤ X < P80), and very good (X ≥ P80). This is especially interesting when the evaluation is done in the healthcare or educational setting, which are essential areas for early problem detection [23].

#### 2.2.2. Peer Relationships

The Classroom Social Experiences Query (CESC) [45], Behaviour and Social Experiences in Class, is a nomination scale designed to assess peer relations in students from 9 to 14 years old. It determines the social status of students, identifying those implicated in behaviours of physical, verbal, and relational abuse, whether in the role of aggressor or victim, as well as the students presenting pro-social behaviours. It includes a sociogram of 12 items corresponding to the constructs of aggression, victimisation, pro-sociality, and sociometric status, according to the model of Coie, Dodge, and Coppotelli [46], which allows for discovery of the rejected, ignored, popular, and controversial students. Part of the translation and adaptation was the Children’s Social Behavior Scale–Peer Report (CSBSP) by Crick and Grotpeter [47], and Children’s Self Experiences Questionnaire–Self Report (CSEQSR) by Crick and Grotpeter [48]. The questionnaires were handled in groups of 20–25 students, where a researcher explained the guidelines for correctly filling it out and informed the students about the anonymous character of the test in order to avoid the tendency of social desirability in some answers. Students were given 20 minutes, which was enough time to answer all the items.

### 2.3. Data Analysis

From 642 participants, 44 cases were discarded because of filling mistakes and/or unfinished PF tests, having a final result of 598 participants (284 males and 314 females). The answer rate was 93.14%. Parametric-type inferential analysis were carried out after checking the normality (Kolmogorov–Smirnov) and homoscedasticity (Levene) assumptions. Results by sex were compared using a Student t-test. The statistic MANOVA was carried out to study the differences in independent variables (PF levels and BMI categories) regarding dependent variables (pro-social behaviours, victimisations, positive status, and negative status). In order to determine the differences among groups, the Bonferroni post-hoc test was carried out. The Pearson bivariate correlations test was carried out to analyse the relations among the different study variables. All the abovementioned statistical procedures were calculated using the Statistical Package for the Social Sciences (SPSSTM), version 24. Significant results were established when the p-value was lower than 0.050.

## 3. Results

Table 1 shows the average percentiles obtained by males and females in each of the PF tests, as well as the number of average nominations on the assessed social behaviours. In CR and SBJ, the percentiles are lower than P50 and there is no difference between males and females (*p* > 0.05), while in HS, the results overcome P50 and the female scores are higher than the male scores (*p* = 0.04). In social behaviours, females were nominated more in pro-sociality (*p* < 0.001) and less in aggressiveness (*p* < 0.001) and negative status (*p* < 0.001), with no differences in the rest of the behaviours.

In Table 2, the correlations between PF component measures (percentiles) and social behaviours (number of nominations) can be noted. Other outstanding features are the positive correlations of CR with pro-sociality (r = 0.220 *p* < 0.01) and positive status (r = 0.277 *p* < 0.01), and negative correlations with negative status (r = −0.156 *p* < 0.01) and victimisation (r = −0.257 *p* < 0.01). The jump is negatively associated with the negative status (r = −0.189 *p* < 0.01) and victimisation (r = −0.302 *p* < 0.01), while the HS is associated negatively with victimisation (r = −0.118 *p* < 0.01) and positively with the positive status (r = 0.214 *p* < 0.01) in the case of males. As for the positive status, it is negatively correlated with BMI (r = −0.199 p < 0.01) and with WC (r = −0.169 *p* < 0.05) only in the case of females. Regarding relations among different PF components, CR and jump are negatively related to the body composition markers (BMC, WC), and HS is positively associated.

Table 3 shows the average scores in the nominations for different levels of PF components. In most of the cases, the highest scores in pro-sociality and positive status were gathered in the highest PF levels; nevertheless, differences were statistically significant in some cases only. Students with a very low level in CR received more nominations in victimisation and negative status, and fewer in pro-sociality and positive status (*p* < 0.001). Likewise, groups with a low or very low level in HS received average scores, which were inferior in positive status (*p* < 0.001) and aggressiveness (*p* = 0.012) Simultaneously, the group with the lowest level in jumping had more nominations in victimisation and in negative status (*p* < 0.001).

Considering that the correlational analysis showed gender differences, the average nominations in different variables of social behaviours were differentiated by gender depending on the weight status of the students classified as underweight, standard weight, overweight, and obese (Table 4). Regarding males, standard-weight students had more nominations in positive status than the rest of their peers, although they were statistically significant regarding the underweight group (*p* = 0.001), as well as in pro-sociality than underweight students (*p* = 0.032). Negative status in underweight students was higher among all the groups (*p* = 0.002). Regarding females, there are outstanding differences in positive status, where the standard weight group had more nominations than the rest of the groups (*p* < 0.001).

## 4. Discussion

The main aim of this study was to analyse the association between PF as a primary indicator of the health state in the physical field and peer relations in children and pre-teens as an indicator of health in the social field. This study found that the group with a lower level of CR obtained more nominations in victimisation, while those groups with a higher level of CR (high and very high) reported higher positive status and more pro-sociality. Furthermore, there were interrelations between the physical state and peer relations, both in pre-teen males and females. CR was the PF component, which was strongly related to positive aspects in peer relations, such as higher pro-sociality and a positive status, and lower victimisation and negative status. Chen [36], studying Asian teenagers, also found that CR was the strongest capacity associated with positive aspects in peer relations, as well as self-esteem. Therefore, CR stands as one of the key components to achieve personal health [49], and is also a critical and concomitant component to improve self-esteem and well-being [50]. 

Muscular fitness was also positively associated, although more circumspectly, to a positive status, and negatively associated with a negative status and victimisation. Hence, students with the lowest level in jumping had more nominations in negative status and victimisation, while a low or very low level in handgrip strength was associated with a lower positive status. These results do not match Chen’s [36], who also found less significant relations of muscular fitness with peer relations and more important relations with self-esteem. Nevertheless, muscular fitness has been found to be related not only with physiological (bone health, injury prevention) and psychological health (self-esteem) found in previous studies [27], but also with social health (better peer relations) in childhood and the teenage years. 

This positive relation between the physical and social state can be explained by different points. On the one hand, the amount of PA is related to an improvement of PF [31], and at the same time, previous research has proven how peers and friends have an important role in the practice of PA. Some studies have shown that physically active children have more friendships and peer relations, and were apparently classified by their peers as popular [14,51]. There are different processes at work, such as friend support and presence, peer standards, friendship quality, and acceptance, which contribute positively to the participation of students in physical activities [16]. Likewise, the environment in which this practice is developed is also important; therefore, relations both with teachers and peers are important regarding motivational increases and enjoyment of practice [13].

Nevertheless, this causal relation between PA and PF is not clear enough. The variation proportion in the different PF measures attributable to PA is weak in children and pre-teens [52], where PF has a strong genetic factor [28].

On the other hand, PF was related to higher general and physical [53], as well as social [54] self-esteem, particularly from the CR component [55]. Accordingly, a better PF is related to a higher perception of competence and PF [56], and these better perceptions contribute to the improvement of general physical health [57,58], as well as social self-esteem [59]. This not only promotes the practice of PA, but also makes peer relations easier. 

In the case of the weight status, associations with different dimensions assessed in peer relations were differentiated by sex. Regarding males, the underweight group showed higher scores in victimisation and negative status and lower scores in positive status. Nevertheless, in females we found a positive correlation between BMI and victimisation, and one which was negative with positive status, so standard-weight females obtained more nominations in positive status than the rest of their peers. Therefore, although some previous studies matched in their results that obese students have more peer relation problems [38], in this study we barely found evidence that supported the fact that pre-teens who are overweight or obese tend to be more victimised or have a more negative social status. Therefore, our results partially match, only in the case of females, with studies such as the one by De la Haye et al. [40], in which it could be observed that those students with obesity received less positive nominations, had a higher risk of victimisation and stigmatisation, and were easily outcast [39,41]. 

Different results found between males and females of this age could have an explanation related to body appearance and the aesthetic models imposed on males and females by society. Body appearance is a concept that has a socio-cultural influence of critical importance to children and pre-teens, which not only has an impact on self-esteem [60], but also determines and is influenced by peer relations [61,62]. Females, in their pre-teen years, start to show higher levels of body dissatisfaction and a greater desire to be thin; thus, a linear relation is established between dissatisfaction and BMI [63]. Nevertheless, children from 10–11 years start to be concerned about their bodily appearance, not only wishing to be thinner, but also bigger and stronger [64]. In contrast to the thin body ideal for females, males in occidental societies are socialised to be more muscular [65]. In fact, some studies with pre-teens and teens found out that the most body-dissatisfied group was the one classified as underweight [66,67,68] These circumstances may explain that thin boys, and therefore, those most far away from the aesthetical model established for males, are less accepted, had a more negative social status or are even more victimised, as the body ideals are determined by conversations and peer relations [62]. This fact is also suitable with the positive correlation found between strength level and a positive status.

Despite the demonstrated results, this study has some limitations that must be highlighted. First of all, it is a cross-sectional study in a particular context, so no causal relations can be established. It is also important to note that this study focused strictly on PF and peer relations, setting aside important variables, such as PA practice, motivation, and other psychological variables that may have been very useful to explain some of the obtained results, and that may be guidelines to work with in future investigations. These results confirm the complexity of the study on the relationships that different health indicators may have, all based on a multi-dimensional understanding of health.

## 5. Conclusions

PF is not only related to benefits in physiological and psychological health, as some previous studies have proved, but it is also related to social health. In this sense, children with better PF maintain better peer relations and higher self-esteem, which has an important role from their childhood and leads to a high and moderate PA. In the case of Spanish pre-teens, PF, and particularly CR, was associated with better peer relations, such as more pro-sociality and positive status and lower victimisation and negative status. Additionally, the weight status was a factor differentiated by sex; while standard-weight females stood out by their positive status, underweight males obtained worse scores in positive status, and lower in negative status—this was all compatible with the aesthetic models of males and females imposed by society, respectively. In school environments, there is a need to develop programmes that promote active lifestyles and focus on the improvement of different PF components, taking into account the influence on children’s social health. These programmes must guarantee moderate and high PA, enough to improve the PF, and they must also rely on pedagogical criteria, providing practice opportunities and a focus on individual improvement that motivates students to participate. Thus, these programmes will not only help to decrease the risk of disease, but also to promote peer relations, self-esteem, and the possibility to create a habit of PA in a friendly, socio-affective environment.

## Figures and Tables

**Table 1 ijerph-17-01890-t001:** Average percentiles in physical fitness (PF) and average nominations in social behaviors. Differences by sex.

Variable	Male (*n* = 284)		Female (*n* = 314)		All (*N* = 598)		
	*M*	*SD*	*M*	*SD*	*M*	*SD*	*p-Value*
CR	40.44	30.96	39.48	31.01	39.93	30.97	0.706
HS	52.25	30.03	57.43	31.31	54.97	30.79	0.040*
SBJ	46.58	35.58	42.38	36.26	44.38	35.97	0.154
BMI	19.21	3.50	19.25	3.47	19.23	3.48	0.898
WC	69.32	9.74	67.20	8.08	68.21	8.96	0.004**
A	9.96	12.63	4.74	7.14	7.22	10.45	0.000***
V	4.81	6.20	4.44	7.57	4.61	6.95	0.520
P	4.34	4.39	5.64	4.61	5.02	4.55	0.000***
PS	5.14	4.07	4.66	3.29	4.88	3.68	0.111
NS	2.73	3.07	1.82	2.41	2.26	2.78	0.000***

* *p* < 0.05; ** *p* < 0.01; *** *p* < 0.001. BMI: body mass index. CR: cardiorespiratory capacity. HS: handgrip strength. SBJ: Standing broad jump. WC: waist circumference. A: aggression. V: victimization. P: prosociality. PS: positive status. NS: negative status.

**Table 2 ijerph-17-01890-t002:** Percentiles correlation between the PF levels and social behaviors.

	BMI	CR	HS	SBJ	WC
CR	−0.353** (♂0.397**♀0.315**)				
HS	0.260** (♂0.290**♀0.238**)	0.023 (♂0.003♀0.043)			
SBJ	−0.140** (♂−0.143*♀−0.138*)	0.393** (♂0.446**♀0.347**)	0.201** (♂−0.066♀0.328**)		
WC	0.676** (♂0.700**♀0.664**)	−0.429** (♂−0.520**♀−0.296**)	0.378** (♂0.480**♀0.260**)	−0.405** (♂−0.520**♀−0.274**)	
A	−0.019 (♂−0.049♀0.019)	0.009 (♂−0.037♀0.074)	0.019 (♂0.066♀0.009)	−0.078 (♂−0.125*♀−0.061)	−0.165** (♂−0.174*♀0.111)
V	−0.016 (♂−0.200**♀0.117*)	−0.257** (♂−0.231**♀−0.279**)	−0.118** (♂−0.122*♀−0.113*)	−0.302** (♂−0.257**♀−0.340**)	0.058 (♂−0.008♀0.124)
P	−0.030 (♂−0.029♀−0.030)	0.220** (♂0.194**♀0.252**)	0.076 (♂0.149*♀0.003)	0.071 (♂0.088♀0.076)	−0.077 (♂−0.143♀−0.042)
PS	−0.112** (♂−0.036♀−0.199**)	0.277** (♂0.225**♀0.337**)	0.112** (♂0.214**♀0.016)	0.086* (♂0.049♀0.120*)	−0.069 (♂0.020♀−0.169*)
NS	−0.052 (♂−0.109♀0.008)	−0.156** (♂−0.133*♀0.195**)	−0.091* (♂−0.028♀−0.137*)	−0.189** (♂−0.210**♀−0.196**)	−0.068 (♂−0.033*♀0.057)

* *p* < 0.05; ** *p* < 0.01; *** *p* < 0.001. BMI: body mass index categorised by IOTF (2012). CR: cardiorespiratory capacity. HS: handgrip strength. SBJ: Standing broad jump. WC: waist circumference. A: aggression. V: victimization. P: prosociality. PS: positive status. NS: negative status.

**Table 3 ijerph-17-01890-t003:** Average scores in the nominations according to the PF and its level. MANOVA and post-hoc.

	Fitness Level												
Variable		Very Low (1)		Low (2)		Medium (3)		High (4)		Very High (5)		MANOVA	Post-Hoc
		*M*	*SD*	*M*	*SD*	*M*	*SD*	*M*	*SD*	*M*	*SD*	*p*	
HS	A	3.76	5.46	6.67	10.29	8.58	11.27	5.21	8.37	7.81	11.39	0.012*	1 < 3,5
	V	5.34	7.51	4.67	7.17	3.96	4.41	3.14	5.91	4.21	6.98	0.254	
	P	3.91	3.90	4.33	3.32	5.76	5.18	5.36	4.59	4.71	4.32	0.050	
	PS	3.19	2.62	3.84	3.05	5.14	2.95	5.97	3.89	4.70	3.64	0.000***	1,2 < 3,4,5
	NS	2.64	2.75	2.15	2.58	2.11	2.45	1.77	2.28	2.26	3.05	0.330	
SBJ	A	8.62	11.22	7.00	11.70	5.82	9.69	6.05	9.78	6.73	9.77	0.235	
	V	8.47	9.73	2.96	3.59	2.32	3.97	2.27	3.26	2.77	4.41	0.000***	1 > 2,3,4,5
	P	4.46	4.30	4.71	4.74	5.79	4.59	4.08	3.67	5.34	4.68	0.081	
	PS	4.40	3.56	4.74	4.05	5.29	3.10	3.95	2.80	5.28	3.78	0.035*	
	NS	3.31	3.53	2.01	2.36	1.20	1.56	1.82	2.18	1.95	2.36	0.000***	1 > 2,3,4,5
CR	A	7.06	10.25	6.51	11.31	3.92	7.03	6.85	9.25	7.71	10.75	0.170	
	V	7.43	9.92	3.75	6.30	2.38	3.79	3.94	4.82	2.23	2.85	0.000***	1 > 2,3,4,5
	P	3.58	3.27	4.48	4.45	6.20	5.33	5.30	5.52	6.32	4.94	0.000***	1 < 3,4,5; 2 < 3,5
	PS	3.50	3.01	4.37	3.59	4.85	3.04	5.57	4.35	6.18	4.01	0.000***	1 < 4,5; 2 < 5
	NS	2.90	3.24	2.29	2.74	1.02	1.41	2.78	3.04	1.55	2.28	0.000***	1 > 3,5; 2,4 > 3,5

* *p* < 0.05; ** *p* < 0.01; *** *p* < 0.001. HS: handgrip strength. SBJ: Standing broad jump. CR: cardiorespiratory capacity. A: aggression. V: victimization. P: prosociality. PS: positive status. NS: negative status.

**Table 4 ijerph-17-01890-t004:** Average scores in the nominations according to the weight status. MANOVA and post-hoc.

		Under-Weight (1)		Normal- Weight (2)		Overweight (3)		Obesity (4)		MANOVA	
		*M*	*SD*	*M*	*SD*	*M*	*SD*	*M*	*SD*	*p*	Post Hoc
Males	A	14.17	16.22	9.72	11.63	10.26	16.64	9.09	12.61	0.517	
	V	7.11	5.84	5.41	6.81	2.07	2.65	2.86	3.77	0.003**	3 < 1,2
	P	2.06	2.85	4.61	4.52	6.19	4.51	2.98	3.50	0.002**	3 > 1,4
	PS	2.17	3.40	5.67	4.21	4.59	3.66	4.32	3.19	0.001**	2 > 1
	NS	5.28	4.50	2.61	2.86	2.33	3.46	2.50	2.64	0.004**	1 > 2,3,4
Females	A	6.44	10.15	4.01	5.66	7.75	10.86	4.29	6.22	0.012*	2 < 3
	V	3.70	5.12	3.98	7.14	4.82	6.33	6.60	10.82	0.190	
	P	4.63	4.00	6.07	4.87	4.20	3.51	5.58	4.34	0.072	
	PS	3.56	2.43	5.41	3.34	3.50	3.10	2.96	2.52	0.000***	2 > 1,3,4
	NS	2.74	3.60	1.60	2.13	2.13	2.84	2.02	2.22	0.087	

* *p* < 0.05; ** *p* < 0.01; *** *p* < 0.001. A: aggression. V: victimization. P: prosociality. PS: positive status. NS: negative status.

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
