# Peer review of "Physical Fitness and Peer Relationships in Spanish Preadolescents"

_ijerph, 2020, doi:10.3390/ijerph17061890_

Round 1

Reviewer 1 Report

Very interesting work

Justify because in the results you are computed calculations with average percentiles

Eta 2 is not described in method.  I suggest removing him from the tables because of his insignificance

The anova must be of repeated measures. Was it done like this?

In general, the manuscript is solid and statistical analysis is appropriate but is neccessary justify because in the results you are computed calculations with average percentiles and The anova must be of repeated measures. Was it done like this?

Author Response

The first we want to do give thanks to the reviewer. All comments have contributed to improve this manuscript, so thanks for it. In the next lines we have answered all comments about the paper. If the reviewer has any question about it, please let we know.

  1. Justify because in the results you are computed calculations with average percentiles.

Thanks for give us this comment. In this way the percentile averages are provided to know the characteristics of the sample and that the results are comparable between them, regardless of age and gender.

  1. Eta 2 is not described in method.I suggest removing him from the tables because of his insignificance.

Done. Thanks for this comment.

  1. The ANOVA must be of repeated measures. Was it done like this?

The statistical MANOVA was carry out (factor: level of fitness in each test: very low to very high) (depend variables: peer nomination in  aggression, victimizations, prosociality, positive status, and negative status). Sorry for the mistake and thanks for the assessment. It has already been modified in the text (P: 4 – L: 163; and Table 3).

In this way and in order to provide more detailed information, Table 4 has been inserted. (P. 6-7; L. 222-224)

  1. In general, the manuscript is solid and statistical analysis is appropriate but is neccessary justify because in the results you are computed calculations with average percentiles and The ANOVA must be of repeated measures. Was it done like this?

Thanks a lot for all comments, they have contributed to improve the manuscript.

Reviewer 2 Report

This manuscript deals with a novel possible connection between physical fitness and relationships between peers. However, in order to move on to the article to the next phase, there are some doubts that must be solved by the authors.

First, the authors claim to have the approval of the bioethics committee, could they provide the approval code?

Secondly, since it was an investigation carried out with minors, was it required to inform the parents or legal guardians of the students? If so, could you attach a copy of the document used? Despite having the approval of the code of ethics, since it is an investigation with minors, it is essential that parents or legal guardians authorize the participation of each subject.

Regarding the battery of physical tests used, why have not the normative values ​​of obesity provided by the WHO been used? Also, why have the results of the eurofit battery been used to analyze the results of a different physical test battery? In addition, line 139 refers to the values ​​of Ortega et al. However, these values ​​are for subjects older than those analyzed in this manuscript.

In relation to omitting the measurement of body fat folds, despite this recommended by the battery itself, could the authors have considered that a higher percentage of body fat would also have an effect on the possible relationships between peers?

Finally, in relation to the questionnaire used, was the questionnaire validated in Spanish pre-adolescent population? If so, could you provide the validation article or its reference?

Regarding formal aspects, please do not include references in the conclusions and check that the bibliographic references conform to the style of the journal.

I consider it a novel article and I am in favor of its publication, but for this, the above issues must be correctly verified.

Author Response

This manuscript deals with a novel possible connection between physical fitness and relationships between peers. However, in order to move on to the article to the next phase, there are some doubts that must be solved by the authors.

The first we want to do give thanks to the reviewer. All comments have contributed to improve this manuscript, so thanks for it. In the next lines we have answered all comments about the paper. If the reviewer has any question about it, please let we know.

1. First, the authors claim to have the approval of the bioethics committee, could they provide the approval code?

We have inserted the code number N202001007 (P. 3; L. 112). Thanks for your comment.

2. Secondly, since it was an investigation carried out with minors,

  1. was it required to inform the parents or legal guardians of the students?

All the participants, parents or legal guardians were informed about the aims, procedures, risks, and benefits of the research and were volunteers among those who provided informed consent. This question was modified in the text (P. 3; L. 103-104). We appreciate this indication.

    b. If so, could you attach a copy of the document used?

Thanks for give us the opportunity to add more information about this. This informed consent comes in Spanish. A copy of the document that was passed to the parents is attached in the last part of the final version of the manuscript. Also we have to said that this study is a part to another big one.

3. Despite having the approval of the code of ethics, since it is an investigation with minors, it is essential that parents or legal guardians authorize the participation of each subject.

Yes. It’s true. We are sorry if in any part of the document that is no clear. We appreciate this comment.

4. Regarding the battery of physical tests used,

a. why have not the normative values ​​of obesity provided by the WHO been used?

We have used IOTF criteria because the values offered by the WHO underestimate the prevalence of overweight and obesity and also because the scientific evidences recommend that they be used. Below are some references that justify it.

M. Kêkê, H.S amouda, J. Jacobs, C. di Pompeo, M. Lemdanid, H. Hubert, D. Zitouni, B. C. Guinhouya (2015). Body mass index and childhood obesity classification systems: A comparison of the French, International Obesity Task Force (IOTF) and World Health Organization (WHO) references. Revue d'Épidémiologie et de Santé Publique, 63(3), 173-182. https://doi.org/10.1016/j.respe.2014.11.003

Bergel, M. L.; Cesani, M. F.; Cordero, M. L.; Navazo, B.; Olmedo, S.; Quintero, F.; Sardi, M.; Torres, M. F.; Aréchiga, J.; Méndez de Pérez, B.; Marrodán, M. D. (2014). Nutritional valuation of schoolchildren from three Ibero-American countries: A comparative analysis of the references proposed by International Obesity Task Force (IOTF) and World Health Organization (WHO). clín. diet. Hosp, 34(1):8-15. DOI: 10.12873/341bergel.

b. Also, why have the results of the eurofit battery been used to analyze the results of a different physical test battery?

We appreciate this comment. But there are no normative values in the Spanish population at these ages, for this reason normative values are used in the European population. In addition, the values used, as well as the protocols, come from tests that are identical in the two batteries.

5. In addition, line 139 refers to the values ​​of Ortega et al. However, these values ​​are for subjects older than those analyzed in this manuscript.

Thanks for this comment. It’s true, the data from that study are in the adolescent population, but only the criteria have been used to establish fitness levels, not percentiles.

6. In relation to omitting the measurement of body fat folds, despite this recommended by the battery itself, could the authors have considered that a higher percentage of body fat would also have an effect on the possible relationships between peers?

Thank you for this indication, since they may be related. However, we have made use of the high priority battery, where the valuation of the folds is not necessary. However, we understand that the indication is very interesting and will be taken into account for future work.

7. Finally, in relation to the questionnaire used,

a. was the questionnaire validated in Spanish pre-adolescent population?

The questionnaire was designed and validated from its origin for pre-adolescent children.

b. If so, could you provide the validation article or its reference?

The reference is indicated in the paper (45) (Collel, Jordi & Escudé, C. Maltrato entre alumnos (I). Presentación d’un cuestionario para evaluar les relaciones entre iguales. CESC Conducta y experiencies sociales a clase. Ámbits Psicopedag. 2006, 18, 8–12.

8. Regarding formal aspects,

a. please do not include references in the conclusions.

Thanks for the comment. The reference was deleted from discussion and bibliographic references sections (P. 8 - L. 302).

b. and check that the bibliographic references conform to the style of the journal.

Thanks for the comment. The bibliographic references were checked and modified.

Round 2

Reviewer 2 Report

Dear authors, many thanks for your accurate comments and for the changes in the manuscript. 

I will give my favorable opinion to publish the manuscript.